# High Frequency of *Cryptosporidium hominis* Infecting Infants Points to A Potential Anthroponotic Transmission in Maputo, Mozambique

**DOI:** 10.3390/pathogens10030293

**Published:** 2021-03-04

**Authors:** Idalécia Cossa-Moiane, Hermínio Cossa, Adilson Fernando Loforte Bauhofer, Jorfélia Chilaúle, Esperança Lourenço Guimarães, Diocreciano Matias Bero, Marta Cassocera, Miguel Bambo, Elda Anapakala, Assucênio Chissaque, Júlia Sambo, Jerónimo Souzinho Langa, Lena Vânia Manhique-Coutinho, Maria Fantinatti, Luis António Lopes-Oliveira, Alda Maria Da-Cruz, Nilsa de Deus

**Affiliations:** 1Instituto Nacional de Saúde (INS), EN1, Bairro da Vila–Parcela n° 3943, Distrito de Marracuene, Maputo 264, Mozambique; adilsonbauhofer@gmail.com (A.F.L.B.); jorfeliachilaule@gmail.com (J.C.); espeguima@yahoo.com.br (E.L.G.); dmbero@gmail.com (D.M.B.); marti.life@hotmail.com (M.C.); bambomiguel@gmail.com (M.B.); elda.muianga07@gmail.com (E.A.); assucenyoo@gmail.com (A.C.); juliassiat@gmail.com (J.S.); je_langa@yahoo.com.br (J.S.L.); lenicouto1@gmail.com (L.V.M.-C.); ndeus1@yahoo.com (N.d.D.); 2Institute of Tropical Medicine, 2000 Antwerp, Belgium; 3Centro de Investigação em Saúde de Manhiça (CISM), Unidade de Pesquisa Social, Manhiça Foundation (Fundação Manhiça, FM), Manhiça 1929, Mozambique; herminiofernando.cossa@gmail.com; 4Instituto de Higiene e Medicina Tropical, Universidade Nova de Lisboa, 1349-008 Lisboa, Portugal; 5Laboratório Interdisciplinar de Pesquisas Médicas, Instituto Oswaldo Cruz-FIOCRUZ, Rio de Janeiro 22040-360, Brazil; maria.fantinatti@gmail.com (M.F.); luizpldeoliveira@gmail.com (L.A.L.-O.); alda@ioc.fiocruz.br (A.M.D.-C.); 6Disciplina de Parasitologia, Faculdade de Ciências Médicas, UERJ/RH, Rio de Janeiro 21040-900, Brazil; 7Departamento de Ciências Biológicas, Universidade Eduardo Mondlane, Maputo 3453, Mozambique

**Keywords:** acute diarrhea, *Cryptosporidium*, children, risk factor, Mozambique

## Abstract

*Cryptosporidium* is one of the most important causes of diarrhea in children less than 2 years of age. In this study, we report the frequency, risk factors and species of *Cryptosporidium* detected by molecular diagnostic methods in children admitted to two public hospitals in Maputo City, Mozambique. We studied 319 patients under the age of five years who were admitted due to diarrhea between April 2015 and February 2016. Single stool samples were examined for the presence of *Cryptosporidium* spp. oocysts, microscopically by using a Modified Ziehl–Neelsen (mZN) staining method and by using Polymerase Chain Reaction and Restriction Fragment Length Polymorphism (PCR-RFLP) technique using 18S ribosomal RNA gene as a target. Overall, 57.7% (184/319) were males, the median age (Interquartile range, IQR) was 11.0 (7–15) months. *Cryptosporidium* spp. oocysts were detected in 11.0% (35/319) by microscopy and in 35.4% (68/192) using PCR-RFLP. The most affected age group were children older than two years, [adjusted odds ratio (aOR): 5.861; 95% confidence interval (CI): 1.532–22.417; *p*-value < 0.05]. Children with illiterate caregivers had higher risk of infection (aOR: 1.688; 95% CI: 1.001–2.845; *p*-value < 0.05). An anthroponotic species *C. hominis* was found in 93.0% (27/29) of samples. Our findings demonstrated that cryptosporidiosis in children with diarrhea might be caused by anthroponomic transmission.

## 1. Introduction

Diarrhea is the one main causes of mortality among children less than 5 years old in low and middle-income countries (LMIC) [1,2]. In Mozambique, it was estimated that 11% of pediatric diseases were due to diarrhea [3]. Rotavirus remains the main etiological agent of diarrhea in children, followed by *Cryptosporidium* spp. [1].

*Cryptosporidium* spp. is an apicomplexan enteric pathogenic parasite protozoan related to water and foodborne outbreaks worldwide [4,5,6,7]. It easily spreads in the environment, through soil, drinking and recreation water (swimming pool, surface waters) or even directly by person-to-person contact and contact with objects surfaces with oocysts [4]. *Cryptosporidium* is also considered an opportunistic parasite that can infect immunocompetent and immunocompromised people [8]. It can also infect wild and domestic animals [4,9,10], which facilitates spread in the environment. 

Currently, more than 40 species of *Cryptosporidium* are recognized, some of which were described recently [9,10]. Even with increased efforts and improved laboratory detection, unknown *Cryptosporidium* species remain to be identified [9]. Among humans, the species most frequently associated with cryptosporidiosis are *C. hominis* (anthroponotic species) and *C. parvum* (anthroponotic and zoonotic species) [9,11,12]. Moreover, *C. parvum* is known as having the broadest range of hosts and is otherwise an important zoonotic species [9,10,11,12].

The Global Enteric Multicenter Study (GEMS) showed that *Cryptosporidium* spp. is the second most attributable pathogen moderate-to-severe diarrhea (MSD) requiring medical attention among young infants [1,13,14]. In Mozambique, several studies demonstrated the presence of this parasite in different populations and/or regions of the country. Additionally, a study in rural Mozambique indicated that *Cryptosporidium* spp. is one of the two pathogens associated with an increased risk of death in children with MSD [15]. Studies of children under 14 years with diarrhea reported different frequencies of *Cryptosporidium* spp. The highest frequency was observed in children aged 0–11 months at 20.0%, followed by 19.0% in children aged 12–23 months, 9.0% in children aged 24–59 months [16], 12.0% in children aged 0–168 months [17] detected by a commercial immunoassay method, and 3.4% to 34.0% in all individuals with diarrhea using microscopy [18,19]. A community study detected *Cryptosporidium* spp. in less than 5.0% of the study population, aged 0 to 48 months old living in poor environment sanitation [20].

Currently, due to its zoonotic and anthroponotic features, it is becoming more important to determine the molecular epidemiology of *Cryptosporidium* spp. Polymerase Chain Reaction (PCR) is a molecular-based method commonly used to study this parasite [10,11,12]. Additionally, this method provides information about the occurrence and distribution of *Cryptosporidium* species, contributing to a better understanding of the parasite. 

Moreover, risk factors also play an important role in infection dynamics. Infection with human immunodeficiency virus (HIV) and consequently development of diarrhea is one of the risk factors that contribute to the chronic diarrhea profile [4,6]. There are few studies in Mozambique that report the risk factors in children with diarrhea and/or that used molecular tools to characterize *Cryptosporidium* [1,15,21]. Altogether, the current analyses were performed with the aim of determining the frequency, risk factors, and molecular diagnostic of *Cryptosporidium* by using a PCR Restriction Fragment Length Polymorphism (PCR-RFLP) in children hospitalized in Maputo City within the context of National Surveillance of Acute Diarrhea (ViNaDiA).

## 2. Results

### 2.1. Characteristics of the Participants

Overall, 319 children were included in the study. A single stool sample was collected from each one. Males composed 57.7% (184/319) of the group. The median age and interquartile interval (in months) were 11 (7–15), with 40.8% (130/319) of children ranging from 7 to 12 months old. Additionally, 38.9% (124/319) of caregivers reported that their children had animal contact, 58.0% (185/319) of the caregivers were literate, and 12.9% (41/319) of the children were HIV-positive (Table 1).

### 2.2. Frequency of Cryptosporidium spp. Infection

During the period of the study, 319 stool samples were collected, examined, and tested for presence of *Cryptosporidium* spp. It was possible to test all 319 samples using the modified Ziehl–Neelsen (mZN) staining method and 192 samples using a PCR method. Only samples with a sufficient stool amount were included for PCR analysis, regardless of results from the staining.

*Cryptosporidium* spp. was detected in 11.0% (35/319) by mZN and in 35.4% (68/192) by PCR (Figure 1). 

As shown in Figure 1, it was not possible to perform molecular analyses on all the 319 samples, because 127 samples had not sufficient quantity for further testing (the remaining 192 samples tested using PCR). In our PCR-based protocol, it was not possible to amplify all genetic material, because 4 positives for mZN failed to amplify. 

### 2.3. Molecular Characterization of Cryptosporidium Species

Of the 192 samples tested by the PCR method, 86.5% (166/192) were negative through the mZN technique, and of those, we were able to recover *Cryptosporidium* DNA in 27.1% (46/166). Overall, 29 samples, including 10.4% (20/192) previously positive and 5.0% (9/192) negative by mZN staining method were successfully genotyped (Figure 1).

A molecular analysis of the 18S rRNA locus identified *C. hominis* in 93.0% (27/29), followed by *C. parvum* in 3.5% (1/29), and mixed infection with *C. hominis* and *C. parvum* in (3.5%, 1/29) (Figure 2). Due to insufficient sample amount, the remaining isolates could not be genotyped.

In the restriction with SspI digestion products for *C. parvum*, *C. hominis* and the mixed infection showed an identical restriction pattern with three visible bands of 111 bp, 254 bp, and 449 bp. A different pattern was seen with AseI digestion products, where *C. parvum* had two visible bands of 104 bp and 268 bp. *C. hominis* also had two visible bands of different molecular sizes, of 104 bp and 561 bp. The mixed infection presented with three bands of 104 bp, 561 bp and 628 bp. 

### 2.4. Risk Factors for Cryptosporidium spp. Infection

Older children were more likely to be infected with *Cryptosporidium* spp. The most susceptible group was children older than two years, compared to younger than seven months (*p*-value < 0.05; adjusted odds ratio (aOR): 5.861, 95% CI: 1.532–22.417) (Table 2). 

Children with illiterate caregivers were more likely to be infected by *Cryptosporidium* spp. than ones with literate caregivers (*p*-value < 0.05; aOR: 1.688, 95% CI: 1.001–2.845) (Table 2).

Being male (crude odds ratio (cOR): 1.252, 95% CI: 0.747–2.097), HIV-positive (cOR: 1.032, 95% CI: 0.466–2.289), and not having animal contact (cOR: 1.280, 95% CI: 0.756–2.166) were not related to infection by *Cryptosporidium* spp. (*p*-value > 0.05) in this study (Table 2).

## 3. Discussion

This study aimed to determine the frequency and risk factors for *Cryptosporidium* spp. infection in children hospitalized in Maputo City through the National Surveillance of Acute Diarrhea (ViNaDiA). 

The frequency of *Cryptosporidium* spp. found in our study was higher than those in studies conducted in different areas (urban, peri-urban and/or rural) and regions from Mozambique (north, south and/or central). Those studies indicate lower frequencies of *Cryptosporidium* spp. if we consider our PCR results (35.4%): 12% in children hospitalized with diarrhea by using ELISA [17], 3.4% in children admitted in pediatric ward in one north central hospital by using mZN and rapid test [18] and 34% in children from ViNaDiA tested before the current study by using mZN, between 2013 and 2015 [19]. The differences in frequencies among these studies can be attributed to different study designs, population characteristics [20,22], diagnostic tools (for instance only microscopy, serology or PCR-based methods) and region of the country (north, south and/or central).

Children older than six months showed higher risk of infection. This trend has been reported in other African countries [23,24,25,26] and also in one of the studies conducted in Mozambique [22]. Higher frequencies in older children (>six months) may be explained by feeding practices, mobility of the child and/or age at which the child has high contact with other children when playing. In Mozambique, exclusive breastfeeding is recommended for the first six months, and then, complementary food is added to the child’s diet. It was previously observed that breastfeeding has a protective effect against any protozoan infection, including *Cryptosporidium* in children from 0 to 48 months old in Maputo, Mozambique [20,27]. The introduction of potentially contaminated complementary foods and the increased mobility of the child expose them to other possible sources of infection, such as soil and animal contact, increasing the risk of infection among older children before mature immunity achieved [4].

Children from illiterate caregivers were more susceptible to infection by *Cryptosporidium* spp. Empirically, literate caregivers suggest that the household is in a higher wealth quintile, compared to those that did not go to school (illiterate). It is reasoned that caregivers who learned better hygiene practices at school are more aware of health risks and practice improved sanitary and hygiene behaviors. 

In our analysis, we found no association between *Cryptosporidium* infection and HIV-status, although more than one third of the children had unknown HIV status. Conversely, in Tanzania [28] and Kenya [25], strong associations between *Cryptosporidium* infection and HIV-status have been observed, with higher frequencies of *Cryptosporidium* infection in HIV-positive children. 

Animal contact was not a predictor for *Cryptosporidium* spp. infection in children. *Cryptosporidium* infection can be acquired through animal contact but can also be transmitted through an anthroponotic route. In our sample, we observed that the majority of the PCR-tested samples contained *C. hominis*. This is commonly reported in Africa, and its acquisition is related to person-to-person transmission [14,21,23,24,28,29], suggesting that animal contact plays a smaller role in infection. 

The increased occurrence of *C. hominis* is corroborated with one hospital-based study of adults with diarrhea in Maputo City which used the 60-kDa glycoprotein gene (gp60) as a target [21]. Occurrence of anthroponomical transmission in children, as described in our study, suggests empiric circulation of the parasite if we consider that adults assist children. An infected adult can easily transmit the parasite to the child, and/or the child can pass it onto another adult or other children.

On the other hand, the PCR-RFLP targeting SSU rRNA (18S rRNA gene) used in this study was the first attempt to molecularly characterize *Cryptosporidium* in children with diarrhea in Mozambique. The target used is the most used among investigators, because this region is less polymorphic, presenting five copies per genome [7,10]. Although gene sequencing could enrich the findings, we did not have the technical conditions at the time of the study. As there was no prior knowledge of the molecular epidemiology of *Cryptosporidium* sp. in our country, we opted for PCR-RFLP to conduct a survey of the circulating species and genotypes. 

Furthermore, we analyzed a single stool samples instead of the optimal multiple (at least 3) consecutive approach, which could result in underestimation. We also applied the PCR technique, which is a highly sensitive diagnostic approach [30,31]. Additionally, this was a hospital-based analysis, meaning that our findings can only be extrapolated to the population from the sites included. However, in four samples mZN-positive, the presence of DNA was not identified. This may have been a consequence of DNA degradation due to suboptimal temperature during transportation and storage, due to the presence of inhibitors [32] or due to a different species with a mutation in the primer’s region [10]. 

There are few studies in Mozambique [1,19], reporting the risk factors for *Cryptosporidium* infection in the children with diarrhea and/or using molecular tools. The findings of this study should receive attention, since the high frequency of *C. hominis* in children observed may be a result of anthroponotic transmission. There is a need to expand the analysis to other provinces of the country and complement it with sequencing tools to better characterize the species in circulation. It is also worth noting that other hosts may be participating in the transmission routes, which is corroborated by the identification of *C. parvum* isolates.

## 4. Materials and Methods 

### 4.1. Ethics Statements

The data used in the present analysis were provided by the ViNaDiA in children. The related protocol was approved by the Mozambique National Bioethics Committee for Health (IRB00002657, reference Nr. 348/CNBS/13). Written informed consent was obtained from children’s parents or legal guardians before questionnaire administration and sample collection.

### 4.2. Study Design, Site and Population

Cross-sectional, hospital-based surveillance was conducted between April 2015 and February 2016 in Hospital Geral de Mavalane (HGM) and Hospital Geral José Macamo (HGJM). These hospitals were selected as sentinel sites because they receive patients from Maputo City and surrounding areas. Both have pediatric out- and inpatient wards. The HGM and HGJM are referral hospitals for both Mavalane and José Macamo health areas and cover fourteen (14) and nine (9) health centers, respectively.

In each sentinel site, focal points (laboratory technicians, physicians and nurses) were identified and trained to screen diarrhea cases, administer the questionnaire, collect and send stool samples to *Instituto Nacional de Saúde (INS)* where the samples were processed. Children up to 60 months old who presented in the sentinel sites with acute diarrhea, defined as three or more loose or liquid stools within 24 h and less than 14 days, were included [33].

### 4.3. Sample Size Calculation

Minimum sample size expected was calculated using OpenEpi [34], with 95% confidence interval (CI), desired precision of 3.0% and an estimated frequency of 4.8% for *Cryptosporidium* spp. from a previous study in children aged up to five years with diarrhea [16]. We obtained a minimum sample size of at least 196.

### 4.4. Data Collection

Demographic, clinical and epidemiological data were assessed by interviewing children’s caregivers with a semi-standardized questionnaire. Information regarding sex, age, animal contact (defined as having physical contact with an animal or their excrements) [16] and caregiver education status were collected. HIV status was self-reported by the children’s caregivers and confirmed in the children’s vaccination cards. If unknown, permission was asked to collect blood samples and tested according to the national testing algorithm. The children newly diagnosed as HIV-positive were followed by the physician at hospital and referred to their neighborhood health facilities for routine assistance after discharge.

### 4.5. Sample Collection and Management

A single stool sample from each child was collected after inclusion. In cases of liquid diarrhea, non-absorbent diapers were used instead of ordinary diapers. Samples were transferred to sterile polystyrene tubes, kept refrigerated in cooler boxes (approximately 2 °C to 8 °C), without preservative and sent to the Laboratory of Parasitology in INS, Maputo. 

A smear was made from fresh stool, and an aliquot was kept in the original tube under 2 to 8 °C for concentration and subsequent microscopic examination for *Cryptosporidium* spp. oocysts. A second aliquot was placed in a vial without preservative, stored under −40 °C and was specifically intended for extracting and purifying genomic deoxyribonucleic acid (DNA) for molecular analysis. A set of previously frozen sub-samples (approximately 0.5 mL) was shipped to *Laboratório Interdisciplinar de Pesquisas Médicas, Instituto Oswaldo Cruz/Fundação Oswaldo Cruz* in Rio de Janeiro, Brazil, under dry ice for DNA extraction, detection and genetic characterization of *Cryptosporidium* species.

### 4.6. Laboratory Sample Processing

#### 4.6.1. Direct Microscopy

The presence of *Cryptosporidium* oocysts in stools was first established using the modified Ziehl–Neelsen (mZN) staining method as described by the World Health Organization (WHO) [35]. Briefly, thin smears of fresh samples and concentrated pellet from formol-ether concentration technique were prepared on the same glass side and air-dried before mZN staining. Samples were read using a microscope; the results were recorded as a positive for those where oocysts of the parasite were visualized under 100X magnification. All stool samples were collected, labeled, processed and stored following the laboratory standard procedures, including Good Laboratory Practice (GLP) recommendations. For the parasitological assay, each internal quality control was made, and all sample readings were double-checked for concordance by two technicians. In case of discordant results, a third observation was required.

#### 4.6.2. DNA Extraction

DNA was extracted from frozen stool sample using a commercial QIAmp stool Mini Kit (Qiagen, Hilden, Germany) following the manufacturer’s protocol with the following modification: the lysis temperature was raised to 95 °C and the DNA was eluted in 100 µL. The pre-treating of the sample and usage of the QIAmp stool Mini Kit was done in a standard procedure as in similar studies, and with this there is no need to freeze or apply thawing cycles [32]. DNA extracts were stored at −20 °C until use.

#### 4.6.3. Molecular Detection by Conventional Polymerase Chain Reaction (PCR)

The 18S rRNA gene was amplified for all samples (positive and negative by mZN staining) by conventional polymerase chain reaction. The forward primer 5’-AACCTGGTTGATCCTGCCAGTAGTC-3’ and reverse primer 5’-TGATCCTTCTGCAGGTTCACCTACG-3’ as described by Xiao et al. [36] were used. Briefly, the PCR contained 10X PCR buffer (MgCl_2_) at a final concentration of 1X, 5mM MgCl_2_, 200 mM (each) deoxynucleotides triphosphate (dNTP), 1.0 U of Taq polymerase (Invitrogen Life Technologies, São Paulo, SP-Brasil), 100 nM (each) primer (Extend, SP-Brasil) and 2.5 µL of DNA template in a total 25 µL reaction mixture. The following parameter was adjusted in our study: MgCl_2_ concentration, Taq polymerase quantity when compared with the study that was used as reference. Each PCR had small adjustments; annealing was set to 61 °C for 45 s and extension to 72 °C for 7 min. The PCR efficacy and the identification of the *Cryptosporidium* genetic material from samples were verified through electrophoresis in agarose gel (1.2%).

Stool samples were considered positive if oocysts with typical characteristics (approximately 4 µm and 6 µm in diameter; stained bright pink within a clear halo under green field) or *Cryptosporidium* DNA with expected base pair (bp) were detected by conventional PCR.

#### 4.6.4. Characterization of *Cryptosporidium* spp. Isolates by Nested PCR and Restriction Fragment Polymorphism (RFLP) Analysis

All samples that were previously positive for 18S rRNA gene amplification were genotyped by a PCR-RFLP technique using genomic DNA as template. Firstly, a PCR product of approximately 1325 bp of the SSU rRNA gene was amplified by a nested PCR using the following primers 5’-TTCTAGAGCTAATACATGCG-3’ and 5’-CCCTAATCCTTCGAAACAGGA-3’ [11,12]. The first PCR contained 1X PCR buffer (MgCl_2_), 6 mM of MgCl_2_, 2.5 U of Taq polymerase, 200 µM of each primer concentration (500 nM each) and 2 µL of DNA template in a total volume of the reaction mixture (50 µL). The initial denaturation was 94 °C for 3 min, followed by the amplification performed in 35 total cycles: 94 °C for 45 s for denaturation, 58 °C for 45 s for annealing and 72 °C for 60 s for extension. The final extension was 72 °C for 7 min. For secondary PCR, for a product of 826 to 864 bp, it was done by using the following primers 5’-GGAAGGGTTGTATTTATTAGATAAAG-3’ and 5’-AAGGAGTAAGGAACAACCTC CA-3’ [11]. A total volume of the reaction mixture (50 µL), Taq DNA amount (1 U) and the primary PCR product was optimized (0.5 µL diluted at 1:20 or 1 µL non-diluted product). Amplification condition for secondary PCR was as follows: 94 °C for 45 s, 59 °C for 30 s and 72 °C for 45 s in 25 cycles with an initial hot start at 94 °C for 3 min and a final extension at 72 °C for 7 min. The PCR efficacy and the identification of the *Cryptosporidium* genetic material from samples were verified through electrophoresis in agarose gel (1.2%).

Genotype identification was made through the analysis of pattern of the secondary product after restriction digestion with the enzymes SspI and AseI (New England BioLabs Inc., Beverly, MA, USA) as described by Xiao et al. [11]. The AseI enzyme has the same digestion function and pattern as the enzyme VspI. Each set of experiments included a negative PCR control (laboratory-grade distilled water). For the restriction, 20 µL of the second product of the nested PCR, 10 U of SspI or AseI, 5 µL pf specific enzyme buffer was digested in a 50 µL of the reaction by 37 °C for one hour under conditions recommended by the supplier manufacturer. Aliquots of amplified and digested fragments were separated and visualized under Ultra-Violet light translucent (Bio-Rad, Milan, Italy) after separation in 1.5% to 2% agarose gel (Invitrogen, Aukland, New Zealand) by electrophoresis stained with 3X GelRed (Biotium, San Francisco, CA, USA). The expected band for each enzyme varied according to the species detected were the most observed highlighted in bold (Table 3).

### 4.7. Data Management and Statistical Procedures

Data were double entered in Epi Info 3.5.1 (CDC, Atlanta, 2008, Atlanta, GA, USA) to minimize entry errors, followed by data comparison and inconsistencies resolution. Data were analyzed using IBM SPSS software (Statistical Package for the Social Science, Armok, NY: IBM Corp, 2011, version 26.0, Chicago, IL, USA). Categorical variables were summarized as frequencies and continuous variables were summarized as medians and Inter-quartile Range (IQR). Contingency tables were built between dependent and independent variables. Crude and adjusted odds ratio were estimated through simple and multiple logistic regression models. Independent variables with *p*-values ≤ 0.2 in the simple logistic regression were included in the multiple logistic regression model in order to obtain adjusted odds ratio. Goodness-of-fit was assesses using the Hosmer and Lemeshow test for the multiple logistic regression model. A p-value less than 0.05 was considered evidence of statistical significance and all analyses were performed considering a 95% confidence interval (95% CI). 

## 5. Conclusions

This study showed a high frequency of *Cryptosporidium* spp. infection detected by PCR-RFLP among children admitted to two public hospitals in Maputo City due to acute diarrhea. Child age (25 to 60 months) and caregiver literacy status were predictors for infection. Our findings demonstrated that the infection is mostly due to *C. hominis*. This suggests mainly anthroponomic transmission, with great public health implications. The present study demonstrated the need for improved public health recommendations by including *Cryptosporidium* spp. in routine testing among children with diarrhea in Mozambique. Moreover, the study presented the need for an establishment of more accurate molecular characterization platform of the parasite in a national context, in order to understand which species occur in the country, routes of infection (if anthroponotic and/or zoonotic) and/or regional patterns, informing public health programs. 

## Figures and Tables

**Figure 1 pathogens-10-00293-f001:**
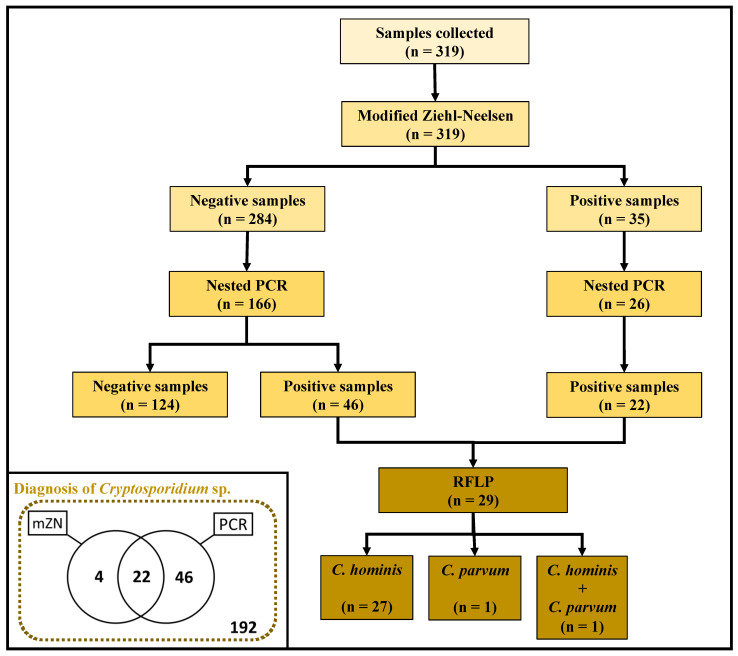
Experimental design. *Cryptosporidium* spp. investigation in samples by microscopy (modified Ziehl–Neelsen staining method) and molecular diagnostic (PCR). The PCR-RFLP was used to *Cryptosporidium* characterization. ZN: modified Ziehl–Neelsen; PCR: polymerase chain reaction; RFLP: restriction fragment length polymorphism.

**Figure 2 pathogens-10-00293-f002:**
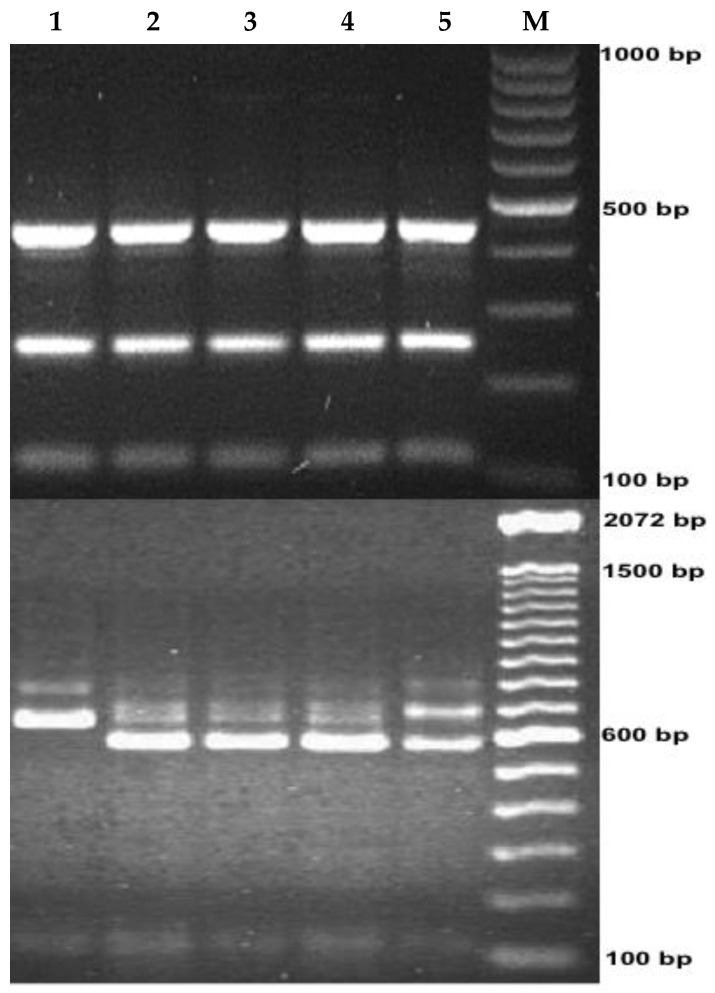
Genotyping of the *Cryptosporidium* parasites by PCR-RFLP targeting 18S rRNA gene. M, molecular size makers (100 bp). Lane 1: *C. parvum*; Lanes 2, 3 and 4: *C. hominis* and Lane 5: mixed infection with *C. hominis* and *C. parvum*. The upper lanes show SspI digestion products showing a molecular size from 111 bp to 449 bp, and the lower lanes show AseI digestion products with molecular size of approximately 104 bp to 628 bp.

**Table 1 pathogens-10-00293-t001:** Demographic and clinical characteristics of the participants enrolled for the study at Hospital Geral de Mavalane (HGM) and Hospital Geral José Macamo (HGJM), Maputo City.

Characteristics	N = 319	Frequency (%)
**Provenience**		
HGM	156	48.9
HGJM	163	51.1
**Sex**		
Female	135	42.3
Male	184	57.7
**Age (in months), categorized**		
0–6	63	19.7
7–12	130	40.8
13–18	81	25.4
19–24	30	9.4
25–60	15	4.7
**Animal contact**		
No	195	61.1
Yes	124	38.9
**Caregiver literacy status**		
Illiterate	133	41.7
Literate	185	58.0
Unknown/missing	1	0.3
**Child HIV status**		
Negative	168	52.7
Positive	41	12.9
Unknown/missing	110	34.5

N = Sample size.

**Table 2 pathogens-10-00293-t002:** Demographic characteristics and animal contact information of children frequencies, crude and adjusted odds ratio for children infected by *Cryptosporidium* spp.

Characteristics	N = 319	n = 81	%	Crude OR (95% CI)	Adjusted OR (95% CI)
**Provenience**					
HGM	156	41	26.3	NA	NA
HGJM	163	40	24.5	NA	NA
**Sex**					
Female	135	31	23.0	1	
Male	184	50	27.2	1.252 (0.747–2.097)	
**Age (in months), categorized**					
0–6	63	6	9.5	1	1
7–12	130	37	28.5	3.780 (1.501–9.517) **	3.604 (1.426–9.112) **
13–18	81	25	30.9	4.241 (1.617–11.124) **	4.170 (1.584–10.979) **
19–24	30	7	23.3	2.891 (0.877–9.533)	2.503 (0.727–8.618)
25–60	15	6	40.0	6.333 (1.671–23.999) **	5.861 (1.532–22.417) **
**Animal contact**					
No	195	53	27.2	1.280 (0.756–2.166)	
Yes	124	28	22.6	1	
**Caregiver literacy status**					
Illiterate	133	42	31.6	1.785 (1.071–2.976) **	1.688 (1.001–2.845) *
Literate	185	38	20.5	1	1
Unknown/missing	1				
**Child HIV status**					
Negative	168	40	23.8	1	
Positive	41	10	24.4	1.032 (0.466–2.289)	
Unknown/missing	110				

N: Total number of samples tested; n: number of positive samples; %: percentage/relative frequency; NA: not applicable; * *p* < 0.05; ** *p* < 0.01.

**Table 3 pathogens-10-00293-t003:** Expected band of the restriction digestion using SspI and AseI enzymes.

Specie	Band Expected (in bp)	
	SspI Digestion	AseI Digestion
*C. hominis*	11, 12,111, 254, 449	70, 102, 104, 561
*C. parvum*	11, 12, 108, 254, 449	102, 104, 628

## Data Availability

The data presented in this study are available on request from the corresponding author. The data are not publicly available due to the ethical reasons.

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
