# Peer review of "High Frequency of Cryptosporidium hominis Infecting Infants Points to A Potential Anthroponotic Transmission in Maputo, Mozambique"

_pathogens, 2021, doi:10.3390/pathogens10030293_

Round 1

Reviewer 1 Report

The second version of the manuscript “High frequency of Cryptosporidium hominis infecting 2 infants points to a potential anthroponotic 3 transmission in Maputo, Mozambique” sent after reviewers’ suggestions is clearer and nice reading.

Minor typing corrections should be considered.

Line 107 : “because 127 samples were had not sufficient” delete were

Line 109 – Consider including a comment for justification of this absence of amplification: false positive or a different species with a mutation in the primer’s region. (or in line 202)

Line 146 – “frequency, and risk factors” delete the comma.

Line 206 – “expand the analysis the analysis to other provinces” delete the repeated words.

Author Response

The second version of the manuscript “High frequency of Cryptosporidium hominis infecting 2 infants points to a potential anthroponotic 3 transmission in Maputo, Mozambique” sent after reviewers’ suggestions is clearer and nice reading.

Response: The authors are thankful for this and appreciate all the efforts made during the revision to improve the manuscript.

Minor typing corrections should be considered.

Line 107: “because 127 samples were had not sufficient” delete were

Response: The authors agreed with the correction and deleted the word “were”. The sentence now is “because 127 samples had not sufficient” (Line 122).

Line 109 – Consider including a comment for justification of this absence of amplification: false positive or a different species with a mutation in the primer’s region. (or in line 202)

Response: The authors are thankful with this suggestion. In the manuscript we had the following sentence related to absence of amplification “This may have been a consequence of DNA degradation due to suboptimal temperature during transportation and storage, due to the presence of inhibitors” (Lines 233 to 235). As suggested, the authors added to the same sentence the following “…or due to a different species with a mutation in the primer’s region” (Lines 235 to 236).

Line 146 – “frequency, and risk factors” delete the comma.

Response: The authors agreed with the correction and deleted the comma the sentence now is “frequency and risk factors” (Line 165).

Line 206 – “expand the analysis the analysis to other provinces” delete the repeated words.

Response: The authors agreed with the correction and deleted the repeated words the sentence now is “expand the analysis to other provinces” (Line 240).

Reviewer 2 Report

Submitted manuscript entitled “High frequency of Cryptosporidium hominis infecting 2 infants points to a potential anthroponotic 3 transmission in Maputo, Mozambique“ (ID: pathogens-1105832) deals with the occurrence of cryptosporidia in children admitted to the hospitals due to diarrhoea. Stool samples of 319 patients (< 5 years old) were examined for the presence of cryptosporidian oocysts using the light microscopy (modified Ziehl-Neelsen staining) and PCR-RFLP technique targeting the 18S ribosomal RNA gene. The PCR-RFLP approach proved to be more sensitive in detection of Cryptosporidium oocysts (35.4%; but due to insufficient stool volume, only 192 samples were analysed) in stool samples than the routine microscopy (11.0%), with children older than two years being most affected. The most frequently found species was the anthroponotic parasite Cryptosporidium hominis (93.0% of the examined samples), followed by C. parvum (3.5%) and mixed infections of both species (3.5%); remaining isolates could not be genotyped due to insufficient sample volume. Authors acknowledge that the analyses of a single stool samples could lead to underestimation. As children with illiterate caregivers were shown to be at higher risk of infection, study concluded that cryptosporidiosis in children with diarrhoea may be caused by anthroponomic transmission.

In general, this study offers original scientific data that is worth publishing. The use of other molecular targets (such as 60-kDa glycoprotein gene) or the inclusion of gene sequencing (e.g., study of Xiao et al. [1999: AEM 3386–3391] showed that sequence analysis of PCR product is needed to differentiate C. wrairi and C. meleagridis from some of the C. parvum genotypes) may be recommended to provide further support for their conclusions, but the authors have already explained their research limitations at the time of study. The authors state that there is no prior knowledge of the molecular epidemiology of Cryptosporidium spp. in their country and only a few studies on cryptosporidiosis in children in Mozambique and the use of molecular detection tools have been published so far. The most important finding appears to be that the high frequency of cryptosporidia in children, in the vast majority represented by C. hominis, may be the result of anthroponotic transmission, which may have a significant impact on public health. Of particular importance is that submitted study points to the urgent need to improve public health recommendations by including Cryptosporidium spp. in routine testing of children with diarrhoea in Mozambique and to better define possible transmission routes.

Taking into account all the circumstances, I have no crucial comments on the presented data. The manuscript, however, would benefit from additional proofreading and review - it seems that the individual sections were written by different authors, reflected in the quality of the text (some examples are given below). Authors should also explain why part of the text is coloured, or unify the text into black, as is customary for submitted manuscripts.

Specific minor comments:

Line 39: Remove comma in “The most affected age group, were children”.

Line 42: “specie C. hominis” change to “species C. hominis

Line 44: “anthroponomical” change to “anthroponomic”

Line 52: “an Apicomplex enteric” change to “an Apicomplexa enteric” (or “an apicomplexan enteric”)

Line 58-59: “Currently, more than 40 species of Cryptosporidium are recognized, some of which were recently 58 [9,10].” This sentence is not completed.

Line 61-63:C. hominis (anthroponotic specie) and C. parvum (anthroponotic and zoonotic specie) [9,11,12]. Moreover, C. parvum is known as having the broadest range of hosts, and is otherwise an important zoonotic specie [9–12].” Correct “specie” to “species”.

Line 64-65: “The Global Enteric Multicenter Study (GEMS) showed that Cryptosporidium spp. is the main 64 reason young infants with moderate-to-severe (MSD) diarrhoea require medical attention [1,13,14].” Reword the sentence.

Line 109: “4 positive” change to “4 positives”

Line 111: Cryptosporidium sp” add a dot after “sp”

Line 170: "literated caregivers" change to "literate caregivers"

Line 171: "iliterated" change to “illiterate“

Line 190: “diarrhea” correct to “diarrhoea”

Line 206: “to expand the analysis the analysis” delete “the analysis”

Line 318: “each enzymes” change to “each enzyme”

Author Response

Recommendation: Language moderate revision

Response: The authors appreciate this recommendations and have now had the manuscript reviewed for language. Additionally, the authors also improved the conclusions supported by the results.

Submitted manuscript entitled “High frequency of Cryptosporidium hominis infecting 2 infants points to a potential anthroponotic 3 transmission in Maputo, Mozambique“ (ID: pathogens-1105832) deals with the occurrence of cryptosporidia in children admitted to the hospitals due to diarrhoea. Stool samples of 319 patients (< 5 years old) were examined for the presence of cryptosporidian oocysts using the light microscopy (modified Ziehl-Neelsen staining) and PCR-RFLP technique targeting the 18S ribosomal RNA gene. The PCR-RFLP approach proved to be more sensitive in detection of Cryptosporidium oocysts (35.4%; but due to insufficient stool volume, only 192 samples were analysed) in stool samples than the routine microscopy (11.0%), with children older than two years being most affected. The most frequently found species was the anthroponotic parasite Cryptosporidium hominis (93.0% of the examined samples), followed by C. parvum (3.5%) and mixed infections of both species (3.5%); remaining isolates could not be genotyped due to insufficient sample volume. Authors acknowledge that the analyses of a single stool samples could lead to underestimation. As children with illiterate caregivers were shown to be at higher risk of infection, study concluded that cryptosporidiosis in children with diarrhoea may be caused by anthroponomic transmission.

In general, this study offers original scientific data that is worth publishing. The use of other molecular targets (such as 60-kDa glycoprotein gene) or the inclusion of gene sequencing (e.g., study of Xiao et al. [1999: AEM 3386–3391] showed that sequence analysis of PCR product is needed to differentiate C. wrairi and C. meleagridis from some of the C. parvum genotypes) may be recommended to provide further support for their conclusions, but the authors have already explained their research limitations at the time of study. The authors state that there is no prior knowledge of the molecular epidemiology of Cryptosporidium spp. in their country and only a few studies on cryptosporidiosis in children in Mozambique and the use of molecular detection tools have been published so far. The most important finding appears to be that the high frequency of cryptosporidia in children, in the vast majority represented by C. hominis, may be the result of anthroponotic transmission, which may have a significant impact on public health. Of particular importance is that submitted study points to the urgent need to improve public health recommendations by including Cryptosporidium spp. in routine testing of children with diarrhoea in Mozambique and to better define possible transmission routes.

Taking into account all the circumstances, I have no crucial comments on the presented data. The manuscript, however, would benefit from additional proofreading and review - it seems that the individual sections were written by different authors, reflected in the quality of the text (some examples are given below). Authors should also explain why part of the text is coloured, or unify the text into black, as is customary for submitted manuscripts.

Response: The authors are thankful for this comment and would like to explain that this is a resubmitted manuscript. The recommendation was to highlight the places where the manuscript was modified, and we used a blue color to show it. At the time, we also added a rebuttal letter explaining the modifications. For the present submission, the authors are highlighting the modifications using track changes. Furthermore, the authors would like to inform that have read the document and harmonized them in order to have the same quality of sentences.

Specific minor comments:

Line 39: Remove comma in “The most affected age group, were children”.

Response: The author agreed with the correction and the sentence now is “The most affected age group were children” (Line 39).

Line 42: “specie C. hominis” change to “species C. hominis

Response: The authors are thankful for this and agreed with the correction and the sentence now was modified for “species C. hominis” (Line 42).

Line 44: “anthroponomical” change to “anthroponomic”

Response: The authors are thankful for this and agreed with the correction and “anthroponomical” was changed to “anthroponomic” (Line 44).

Line 52: “an Apicomplex enteric” change to “an Apicomplexa enteric” (or “an apicomplexan enteric”)

Response: The authors agreed with the correction and the sentence now has “an apicomplexan enteric” (Line 58).

Line 58-59: “Currently, more than 40 species of Cryptosporidium are recognized, some of which were recently 58 [9,10].” This sentence is not completed.

Response: The authors are thankful for this, and added “described” the sentence now is “Currently, more than 40 species of Cryptosporidium are recognized, some of which were described recently” (Line 65 to 66).

Line 61-63: “C. hominis (anthroponotic specie) and C. parvum (anthroponotic and zoonotic specie) [9,11,12]. Moreover, C. parvum is known as having the broadest range of hosts, and is otherwise an important zoonotic specie [9–12].” Correct “specie” to “species”.

Response: The authors agreed with the correction and wrote species instead of specie. The sentence now is “C. hominis (anthroponotic species) and C. parvum (anthroponotic and zoonotic species). Moreover, C. parvum is known as having the broadest range of hosts, and is otherwise an important zoonotic species” (Lines 67 to 70).

Line 64-65: “The Global Enteric Multicenter Study (GEMS) showed that Cryptosporidium spp. is the main 64 reason young infants with moderate-to-severe (MSD) diarrhoea require medical attention [1,13,14].” Reword the sentence.

Response: The authors are thankful for this comment and reworded the sentence. The sentence now is “The Global Enteric Multicenter Study (GEMS) showed that Cryptosporidium spp. is the second most attributable pathogen moderate-to-severe diarrhoea (MSD) requiring medical attention among young infants” (Line 71 to 73).

Line 109: “4 positive” change to “4 positives”

Response: The authors agreed with the correction and changed “positive” to “positives”, the sentence now is “4 positives” (Line 124).

Line 111: Cryptosporidium sp” add a dot after “sp”

Response: The authors are thankful and added a dot after “sp.” (Line 130).

Line 170: "literated caregivers" change to "literate caregivers"

Response: The authors appreciates this comment and change "literated caregivers" to  "literate caregivers" (Line 189).

Line 171: "iliterated" change to “illiterate“

Response: The authors appreciate this comment and change "iliterated" to  "iliterate" (Line 190).

Line 190: “diarrhea” correct to “diarrhoea”

Response: The authors agreed with this correction and wrote “diarrhoea” (Line 223).

Line 206: “to expand the analysis the analysis” delete “the analysis”

Response: The authors are thankful for this comments and deleted the repeated word  “the analysis” the sentence now is “to expand the analysis” (Line 240).

Line 318: “each enzymes” change to “each enzyme”

Response: The authors are thankful for this comment and changed “each enzymes” to “each enzyme”(Line 369).

This manuscript is a resubmission of an earlier submission. The following is a list of the peer review reports and author responses from that submission.

Round 1

Reviewer 1 Report

The manuscript “High frequency of Cryptosporidium hominis infecting 2 infants points to a potential anthroponotic 3 transmission in Maputo, Mozambique” aims to report the frequency, the risk factors, and the genetic diversity of Cryptosporidium species in children admitted in two public hospitals in Maputo city, Mozambique.

It is a well written manuscript in one important theme for research in neglected tropical diseases.

In general, it is a good manuscript, however several changes should be performed through the manuscript to be suitable.

First, the aim to determine “genetic diversity” should be deleted since the author just use molecular diagnose techniques and have no results concerning genetic diversity. Moreover, all the Material and methods sub chapter relating to molecular detection should be rewrite.

Specific changes:

Results:

Lin 99 – Please explain the method used to select samples for PCR (68/192). Just the positives?  

Line 101 and 102 sentence “On the other side, using PCR-based protocol, it was not possible to amplify” – Sentence not clear. Please clarify the number of samples without enough sample and with failure PCR  

Figure 1 The outside branch between “Tested by PCR-RFLP” and “positive for Cryptosdporidium” is not sufficiently clear because it is not indicated which ones were PCR negative.

Figure 2 – please explain the bands and the expected size in the text and in the legend.

Discussion:

Line 132 (and 179). Instead of referring the sample was “genetically characterize”, I would prefer molecular diagnose, since PCR-RFLP just differentiate species. (the same in line 179)

Line 142-144 “The differences among studies can be due to different studies designs, population characteristics, and diagnosis tools, including the target loci (in case of PCR-based methods) and region of the country (north, south and/or center)” Please explain how the selected loci can influence the results of the different studies. The region of the country is the same as “population characteristics”.

I would suggest the inclusion of a sentence with final considerations about the importance of this study to Mozambique.

Materials and Methods

In subchapter 4.4 please refer if the children newly diagnose for HIV was included in follow up consultations

Line 253-254 “The following parameter was optimized: MgCl2 concentration, Taq polymerase quantity”. Please clarify this sentence. Optimize from what?? I would suggest including the used parameters.

In line 255 the authors refer a “protocol previously described” but it is the first time they refer the protocol. Once again include the used parameters

Table 3 – I would suggest deleting this table because it does not include new information.

Line 260 to 262- I would suggest including the microscopy information in the subchapter 4.6.1

Line 266. “technique using genomic DNA as template rather than PCR product” This sentence is very strange. Please exclude or explain.

Line 267-268 – Please refer which loci the referred primers amplify.

Line 269-270 “The following parameter was optimized for the first PCR” Instead of saying that was optimized please refer the used conditions to be repeated by others.

Lines 279-285 – Please refer all the protocol including PCR bands expected and sizes of bands after enzyme restriction.

Line 281 – Once again there is reference a previously described protocol, but I have not found it.  

Author Response

Comments and Suggestions for Authors

The manuscript “High frequency of Cryptosporidium hominis infecting 2 infants points to a potential anthroponotic 3 transmission in Maputo, Mozambique” aims to report the frequency, the risk factors, and the genetic diversity of Cryptosporidium species in children admitted in two public hospitals in Maputo city, Mozambique.

It is a well written manuscript in one important theme for research in neglected tropical diseases.

In general, it is a good manuscript, however several changes should be performed through the manuscript to be suitable.

First, the aim to determine “genetic diversity” should be deleted since the author just use molecular diagnose techniques and have no results concerning genetic diversity. Moreover, all the Material and methods sub chapter relating to molecular detection should be rewrite.

Response: The authors changed the manuscript accordingly. Genetic diversity was changed to molecular diagnose. Also, the Material and methods sub chapter’s 4.6.3 and 4.6.3 related to molecular detection was rewritten (Lines 607 to 684).

Specific changes:

Results:

Lin 99 – Please explain the method used to select samples for PCR (68/192). Just the positives?

Response: All samples were first analyzed using a microscopy diagnostic technique. Those with sufficient amount were submitted to PCR regardless of the positive/negative result by microscopy. Following the reviewer comments, the sentence in the sub chapter 2.2 referring which samples were used for PCR was added and now is written “Only samples with a sufficient stool amount were included for PCR analysis, regardless of results from the staining” (Lines 223 to 225).

Line 101 and 102 sentence “On the other side, using PCR-based protocol, it was not possible to amplify” – Sentence not clear. Please clarify the number of samples without enough sample and with failure PCR  

Response: Of the 319, 192 samples had sufficient stool amount for molecular diagnose, of those four failed to amplify, the full sentence in the sub chapter 2.2 it is now written as followed “As shown in Figure 1, it was not possible to perform molecular analyses on all the 319 samples, because 127 samples were had not sufficient quantity for further testing (remaining the 192 samples tested for PCR). In our PCR-based protocol, it was not possible to amplify all genetic material, because 4 positive for mZN failed to amplify” (Lines 228 to 231).

Figure 1 The outside branch between “Tested by PCR-RFLP” and “positive for Cryptosdporidium” is not sufficiently clear because it is not indicated which ones were PCR negative.

Response: The authors are thankful for this point, and the figure was adjusted by adding information about the number of positive samples by mZN used in the PCR method.

Figure 2 – please explain the bands and the expected size in the text and in the legend.

Response: Bands and expected size were explained in the text the sentence is now as following “In the restriction with SspI digestion products for C. parvum, C. hominis and the mixed infection showed an identical restriction pattern with three visible bands of 111 bp, 254 bp, and 449 bp. A different pattern was seen with AseI digestion products, where C. parvum had two visible bands of 104 bp and 268 bp. C. hominis also had two visible bands of different molecular sizes, of 104 bp and 561 bp. The mixed infection presented with three bands of 104 bp, 561 bp, and 628 bp”  (Lines 264 to 268) and in the legend of figure 2 as following “…The upper lanes show SspI digestion products with molecular size from 111 bp to 449 bp and the lower lanes show AseI digestion products with molecular size from 104 bp to 628 bp” (Lines 294 to 296).

Discussion:

Line 132 (and 179). Instead of referring the sample was “genetically characterize”, I would prefer molecular diagnose, since PCR-RFLP just differentiate species. (the same in line 179)

Response: The authors agree with this recommendation, and used the expression “molecular diagnostic” instead of “genetically characterize” in all the relevant text.

Line 142-144 “The differences among studies can be due to different studies designs, population characteristics, and diagnosis tools, including the target loci (in case of PCR-based methods) and region of the country (north, south and/or center)” Please explain how the selected loci can influence the results of the different studies. The region of the country is the same as “population characteristics”.

Response: The authors are thankful for this comment. The above statement was meant to indicate that the diagnostic method could have an impact of the true estimate of the parasite occurrence as each may have different sensitivity and specificity. The expression “loci” was removed, and the sentence integrates the usage of different diagnostic tools, the sentence is now written as following “The differences in frequencies among these studies can be attributed to different study designs, population characteristics, diagnostic tools (for instance only microscopy, serology or PCR-based methods), and region of the country (north, south and/or central)” (Lines 338 to 340).

I would suggest the inclusion of a sentence with final considerations about the importance of this study to Mozambique.

Response: Following the Reviewer the authors has added a sentence with final consideration about the importance of this study to Mozambique in chapter 5, with the following “…The present study demonstrated the need for improved public health recommendations by including Cryptosporidium spp. in routine testing among individuals with diarrhoea in all provinces of Mozambique. Also, the study presented the need for an established molecular characterization platform of the parasite in a national context, in order to understand which species occur in the country, their modes of infection (if anthroponotic and/or zoonotic), and/or regional patterns, informing public health programs” (Lines 701 to 707).

Materials and Methods

In subchapter 4.4 please refer if the children newly diagnose for HIV was included in follow up consultations

Response: Newly diagnosed children were followed by the physician at hospital and referred to their neighborhood health facilities for routine follow up, the sentence added is as following “The children newly diagnosed as HIV-positive were followed by the physician at hospital, and referred to their neighborhood health facilities for routine assistance after discharge” (Lines 560 to 562).

Line 253-254 “The following parameter was optimized: MgCl2 concentration, Taq polymerase quantity”. Please clarify this sentence. Optimize from what?? I would suggest including the used parameters.

Response: The authors intended to indicate that adjustments in the reagent amounts/concentrations were made for the processing. Due to this, the authors clarified in the manuscript by using the expression “adjusted” instead of optimized, and noted that this was in comparison to the original protocol by Xiao et al.(1999) (Lines 615 and 619).

In line 255 the authors refer a “protocol previously described” but it is the first time they refer the protocol. Once again include the used parameters

Response: The authors added the information regarding the reference of the protocol used to perform the PCR (Line 615).

Table 3 – I would suggest deleting this table because it does not include new information.

Response: Following reviewer recommendations, Table 3 was removed.

Line 260 to 262- I would suggest including the microscopy information in the subchapter 4.6.1

Response: Microscopy information was moved to sub chapter 4.6.1 and the sentence added is as followed “Samples were read using a microscope, the results were recorded as a positive or negative for those where oocysts of the parasite were visualized under 100X magnification. All stool samples were collected, labeled, processed and stored following the laboratory standard procedures, including Good Laboratory Practice (GLP) recommendations. For the parasitological assay, each internal quality control was made and all sample readings were double-checked for concordance by two technicians. In case of discordant results, a third observation was required” (Lines 581 to 587).

Line 266. “technique using genomic DNA as template rather than PCR product” This sentence is very strange. Please exclude or explain.

Response: The authors are thankful for this observation and removed the expression “rather than PCR product” (Line 631).

Line 267-268 – Please refer which loci the referred primers amplify.

Response: The authors added information about the loci (gene) referred for primer amplification in sub chapter 4.6.4 (Line 632).

Line 269-270 “The following parameter was optimized for the first PCR” Instead of saying that was optimized please refer the used conditions to be repeated by others.

Response: The authors are thankful with this point and the text was revised. The PCR conditions were added in detail in sub chapter 4.6.4 and the sentence is now written as followed “The first PCR contained 1X PCR buffer (MgCl2), 6 mM of MgCl2, 2.5 U of Taq polymerase, 200 µM of each primer concentration (500 nM each), 2 µL of DNA template in a total volume of the reaction mixture (50 µL). The initial denaturation was 94oC for 3 minutes, followed by the amplification performed in 35 total cycles: 94oC for 45 seconds for denaturation, 58oC for 45 seconds for annealing and 72oC for 60 seconds for extension. The final extension was 72oC for 7 minutes” (Lines 633 to 638).

Lines 279-285 – Please refer all the protocol including PCR bands expected and sizes of bands after enzyme restriction.

Response: The authors added the detailed protocol to enzyme restriction in sub chapter 4.6.4 and the sentence is added is “Each set of experiments included a negative PCR control (laboratory-grade distilled water). For the restriction, 20 µL of the second product of the nested PCR, 10 U of SspI or AseI, 5 µL of specific enzyme buffer was digested in a 50 µL of the reaction by 37 oC for one hour under conditions recommended by the supplier manufacturer“ (Lines 650 to 653). Additionally, the authors added in sub chapter 4.6.4 a table 3 that describes the expected band size by the enzyme (Lines 683 to 684).

Line 281 – Once again there is reference a previously described protocol, but I have not found it.  

Response: The authors are thankful with this comment, and added the reference of the protocol used for the restriction (Line 649).

Reviewer 2 Report

This is a well-written article presenting data on the frequency and risk factors of Cryptosporidium infection in children with diarrhea in the city of Maputo in Mozambique. In addition, Cryptosporidium isolates are characterized by RFLP PCR.

Comments:

  1. Cryptosporidium DNA extraction method used in this work is not the optimum one. The high resistance of Cryptosporidium oocysts makes it necessary to use a more aggressive method of breaking the oocysts that achieves a higher extraction performance. The best methods for the isolation of DNA of these protozoa use several cycles of flash freezing the sample in liquid nitrogen followed by rapid thawing in a boiling water bath followed by treatment with proteinase K, or with commercial kits using cell disruptors. Due to this fact, in this work positive by staining samples were not positive by PCR.
  2. On the other hand, the authors should have only performed the nested PCR described in the article both for the detection of positive samples for Cryptosporidium and for the characterization of the isolates. They would have saved a step (the conventional PCR) and also, they would surely have obtained a greater number of positive samples, since with the nested PCR the sensitivity is considerably increased.
  3. The authors used the restriction enzymes SspI and AseI in the RFLP assay. However, according to the articles that are referenced, the enzymes used are SspI for the differentiation of the different Cryptosporidium species except for C. parvum and C. hominis. To differentiate these two species, it would be necessary to use VspI and not AseI, that they used.

Author Response

Recommendation: Language extensive revision

Response: The authors appreciate this recommendations and have now had the manuscript reviewed for language.

Comments and Suggestions for Authors

This is a well-written article presenting data on the frequency and risk factors of Cryptosporidium infection in children with diarrhea in the city of Maputo in Mozambique. In addition, Cryptosporidium isolates are characterized by RFLP PCR.

Comments:

Cryptosporidium DNA extraction method used in this work is not the optimum one. The high resistance of Cryptosporidium oocysts makes it necessary to use a more aggressive method of breaking the oocysts that achieves a higher extraction performance. The best methods for the isolation of DNA of these protozoa use several cycles of flash freezing the sample in liquid nitrogen followed by rapid thawing in a boiling water bath followed by treatment with proteinase K, or with commercial kits using cell disruptors. Due to this fact, in this work positive by staining samples were not positive by PCR.

Response: In fact, Cryptosporidiumhave highly resistant oocysts and because of that it is challenging to break them for a successful DNA extraction. We have used the Qiagen kit and, due to the resistance of the oocyst wall, the lysis temperature was from 70°C to 95°C. The authors are thankful for the suggestion about the usages of liquid nitrogen, term shocking and treatment with proteinase K to increase the positivity of the parasite and will take it into account in further molecular studies.

On the other hand, the authors should have only performed the nested PCR described in the article both for the detection of positive samples for Cryptosporidium and for the characterization of the isolates. They would have saved a step (the conventional PCR) and also, they would surely have obtained a greater number of positive samples, since with the nested PCR the sensitivity is considerably increased.

Response: The authors acknowledge this suggestion and will take it into consideration in future analyses.

The authors used the restriction enzymes SspI and AseI in the RFLP assay. However, according to the articles that are referenced, the enzymes used are SspI for the differentiation of the different Cryptosporidium species except for C. parvum and C. hominis. To differentiate these two species, it would be necessary to use VspI and not AseI, that they used.

Response: In papers referenced in our manuscript, the enzyme VspI was used instead of AseI for restriction purposes. The enzyme AseI has the same pattern and performance as VspI [1]. To avoid misunderstanding, the authors added the information regarding the same resolution of the two enzymes in the sub chapter 4.6.4 (Lines 649 to 650).

Reference

  1. VspI (AseI) (10 U/ΜL) Available online: https://www.thermofisher.com/order/catalog/product/ER0911 (accessed on 26 December 2020).

Round 2

Reviewer 2 Report

Dear authors, 

This manuscript is of interest, however the methods used are not appropriate for a high index journal. Also, it should be a short report, as the results are not sufficient for a regular manuscript.

Kindest regards.